# Effect of Epoxidized and Maleinized Corn Oil on Properties of Polylactic Acid (PLA) and Polyhydroxybutyrate (PHB) Blend

**DOI:** 10.3390/polym14194205

**Published:** 2022-10-07

**Authors:** Jaume Sempere-Torregrosa, Jose Miguel Ferri, Harrison de la Rosa-Ramírez, Cristina Pavon, Maria Dolores Samper

**Affiliations:** Instituto de Tecnología de Materiales (ITM), Universitat Politècnica de València (UPV), Plaza Ferrándiz y Carbonell 1, 03801 Alcoy, Alicante, Spain

**Keywords:** polylactic acid, polyhydroxybutyrate, blend, modified corn oil

## Abstract

The present work analyzes the influence of modified, epoxidized and maleinized corn oil as a plasticizing and/or compatibilizing agent in the PLA–PHB blend (75% PLA and 25% PHB wt.%). The chemical modification processes of corn oil were successfully carried out and different quantities were used, between 0 and 10% wt.%. The different blends obtained were characterized by thermal, mechanical, morphological, and disintegration tests under composting conditions. It was observed that to achieve the same plasticizing effect, less maleinized corn oil (MCO) is needed than epoxidized corn oil (ECO). Both oils improve the ductile properties of the PLA–PHB blend, such as elongation at break and impact absorb energy, however, the strength properties decrease. The ones that show the highest ductility values are those that contain 10% ECO and 5% MCO, improving the elongation of the break of the PLA–PHB blend by more than 400% and by more than 800% for the sample PLA.

## 1. Introduction

Nowadays, polymeric materials have replaced a significant part of the different materials employed in the engineering sector [1], as well as other sectors such as construction [2], key parts in electronics appliances [3,4,5], and food packaging [6,7]. Consequently, the demand for these resources has considerably increased in recent years [8]. 

These resources, mostly of petrochemical origin, are obtained from non-renewable and limited sources which increase their depletion and the price of energy resources [9]. Additionally, the production of these resources has a devastating impact on our planet, causing environmental problems and large amounts of CO_2_ emissions which directly increase global warming and destroy ecosystems [10]. Another problem derived from the materials of petrochemical origin is the accumulation of plastic residues that mostly end up in landfills or in the oceans, causing microplastic migrations that contaminate and spoil wildlife [11,12]. The strategies and directives proposed by the European Commission give priority to sustainable and non-toxic single-use products, aiming first and foremost to reduce the quantity of plastic waste [13]. 

Therefore, one conceivable solution for this massive problem is the recycling and reuse of these materials with the aim of using less virgin material [8]. This is a good way to fight against this problem, however, today, this is not enough and another solution is required. Another possible way to reduce the number of materials of petrochemical origin would be to increase the use of biodegradable materials. Nevertheless, polymers still need research to improve their properties and facilitate their manufacture for them be comparable to other polymers of petrochemical origin [14,15].

One of the most widely used biodegradable materials today is polylactic acid (PLA) [16,17]. PLA is a linear aliphatic polyester which is produced from a fermentation process of corn starch, wheat starch, among others, meaning that it comes from renewable resources [18,19]. Considering the outstanding properties of PLA such as biocompatibility and high strength, it also has some slight disadvantages such as its low ductility which reduce its possible applications [20,21,22].

Due to the brittleness of PLA, it is often modified with other polymers: thanks to the compatibility of PLA, it can be blended with other polymers to improve its properties. In addition, such a combination can also have other benefits for both polymers, so it is interesting to make such modifications. Some of the petroleum-based materials with which PLA has good miscibility are polyethylene terephthalate (PET) [23,24], polyvinyl chloride (PVC) [25], thermoplastic elastomers (TPEs) [26,27], polypropylene (PP) [28,29] and polyethylene (PE) [30,31,32]—all of which are commodity plastics. 

Because of the aforementioned problems, biopolymers for blending with PLA such as polyhydroxyalkanoates (PHAs) [33,34,35], poly (E-caprolactone) (PCL) [36,37,38], or polybutylene adipate-co-terephthalate (PBAT) [39,40,41] have been tested and can achieve improvements in mechanical properties or degradation stability. 

Plasticizers, low-molecular-weight molecules that generally improve the ductility and processing of polymers, can be used to improve the miscibility between materials. There are two types of plasticizers, depending on their origin, namely petrochemical, such as tar pitch polymerization resins, etc. Some of these are used because they are environmentally friendly and non-toxic plasticizers, such as poly (ethylene glycol) (PEG) [42,43,44,45,46], poly (propylene glycol) (PPG) [47], lactic acid oligomer (OLA) [48], triethyl citrate (TEC) [49], and tributyl citrate (TBC) [50,51,52]. In addition, a second option of natural origin is vegetable and animal oils, acids, etc. Some of the most commonly used are epoxidized palm oil (EPO) [53], linseed oil (ELO) [54], octyl epoxy stearate (OES) [55], karanja oil (EKO) [56], epoxidized soybean oil (ESBO) [53], or epoxidized cottonseed oil (ECSO) [57]. 

Overall due to PLA’s compatibility, it can be blended with other polymers such as poly-B-hydroxybutyrate (PHB) [58,59,60], among other starch acetylated thermoplastics discussed above, to improved its properties and achieve significant improvements in the mechanical properties or the degradation stability.

Thus, the aim of the work was to improve the properties of PLA by blending it with PHB at 75–25%, respectively, to improve the miscibility and reduce the melting temperatures to promote its use in certain industries. For this purpose, different percentages of epoxidized corn oil and maleinized corn oil, 1, 2.5, 5, 7.5, and 10 wt%, were used as plasticizers. The processes of the maleinization and epoxidation of vegetable oils provide greater reactivity to the oil and greater thermal stability with respect to unmodified oils. Therefore, the compatibility of the oils with PLA is improved and the plasticizing effect is increased [56,61]. These oils were modified from corn oil due to certain characteristics that make it interesting compared to other oils from cereals or seeds. Corn oil is interesting since it has an interesting lipid profile for functionalization and has the advantage, compared to other modifiable oils, of being able to be produced in large quantities and in many countries of the world, as it is economic and very stable to oxidation. Its good oxidative stability is related to its low α-linolenic acid content, which is the case of corn oil between 0–2% and depends on the genotype [62]. Among the fatty acids contained in the oil, 85% are unsaturated, which means a significant amount of double bonds that allow its functionalization by chemical processes such as epoxidation and maleinization. Therefore, this work evaluates the potential of epoxidized corn oil (ECO) and maleinized corn oil (MCO) as bio-based plasticizers to improve the ductile properties of PLA–PHB blends.

## 2. Materials and Methods

### 2.1. Materials

Poly (lactic acid), PLA commercial grade Biopolymer 2003D, was supplied by Nature Works LLC (Minnetonka, MN, USA) in pellet form with a density of 1.24 g cm^−3^. The PLA–PHB blend containing 25% PHB was supplied by Biomer (Krailling, Germany) in pellet form with a density of 1.25 g cm^−3^ with commercial grade P226E. The vegetable oil used to carry out the epoxidation and maleinization process was food-grade corn vegetable oil (CO—corn oil).

### 2.2. Corn Oil Modification

The corn oil modification processes were carried out in a three-necked round bottom flask with a capacity of 500 mL in order to incorporate all the necessary elements. One of them was used to introduce the reagents, another one for temperature control and the central one to connect the reflux condenser.

#### 2.2.1. Epoxidation Process

The epoxidation process started by adding 182.0 g of corn oil into the flask until 50–55 °C was reached with magnetic stirring. Then, 18.64 mL of acetic acid was added until the temperature was stable at 60 °C, then a mixture of 182.70 mL of hydrogen peroxide (3:1 M ratio (peroxide:oil)) with 1.46 mL of sulfuric acid was added dropwise, and the slow addition of this mixture was performed to avoid an exothermic reaction and with the heating mantle off. Once the addition of this mixture was finished and the reaction was controlled, the working temperature was increased between 80 and 85 °C for 8 h. Samples were collected at each hour of the epoxidation process according to ASTMD1652-97 [63] and ISO 3961:2009 [64] oxirane oxygen methods. Prior to the test, the samples were washed with distilled water and purified for 30 min in a centrifuge at 4000 rpm. Figure 1 shows a schematic representation of the epoxidation process.

#### 2.2.2. Maleinization Process

The maleinization ratio used was 2.4:1, following the recommendations of previous works [65,66]. A total of 27 g of maleic anhydrive (MA) and 300 g of corn oil were used, the maleinization process was carried out in three stages at different temperatures—namely 180, 200, and 220 °C—and the samples were taken for acid value (AV) calculations every 60 min. Initially, the oil was introduced in the round flask and when it reached the temperature of 180 °C, 9 g of MA (1/3 of the total) was added. The temperature of 180 °C was maintained for one hour and increased to 200 °C, before another 9 g of MA was added and maintained again for 1 h and the same process was carried out at 220 °C. Finally, the mixture was cooled to room temperature. The degree of maleinization was determined following the guidelines of ISO 660:2009 [67] and using the following expression:(1)$$Acid\;Value=\frac{56.1\;\cdot \;c\;\cdot \;V}{m}$$
where *c* is the exact concentration of the KOH standard solution used (mol-L^−1^), *V* is the volume of the KOH standard solution used (mL), and *m* is the analyzed mass (g). A schematic representation of the maleinization process can be seen in Figure 1.

### 2.3. Sample Preparation

The samples used in the different experimental techniques and tests were obtained by mixing processes in the laboratory. Before handling the PLA and PHB, both materials were dried at 60 °C for 24 h to eliminate the percentage of moisture indicated by the supplier.

First, different mixtures were made with 75% PLA, 25% PHB and different percentages of epoxidized and maleinized corn oil, as shown in Table 1. The mixtures were left to dry for a minimum of 24 h at 60 °C.

To achieve homogeneity in the mixtures, an extrusion process was carried out. The temperature range program used for the different zones of the extruder was the following: 175–180–180–185 °C, from the hopper to the nozzle, respectively, at constant speed of 40 rpm. At the end of the extrusion process, the material was cut until it was reduced to pellet size.

Subsequently, the material in the pellets form was injected to obtain standard specimens for subsequent characterization. The chosen temperature program was 175–180–185 °C, from the hopper to the nozzle, respectively. The temperatures used in the manipulation process were those recommended by the material supplier.

### 2.4. Characterization Techniques

#### 2.4.1. Mechanical Properties

The mechanical characterization was carried out with tensile, flexural, impact, and hardness tests. The tensile and flexural tests were performed on an ELIB 30 universal testing machine (S.A.E. Ibertest, Madrid, Spain). A minimum of five different specimens were tested using a 5 KN load cell. The speed used in the tests was 5 mm min^−1^, at room temperature. For the tensile test, the parameters defined in the ISO 527-1:2019 standard [68] were followed, and for the flexural test, the ISO 178:2019 standard [69] was used. In addition, from the integral of the stress–strain curves obtained in the tensile tests, the toughness modulus was calculated.

To determine the amount of energy absorbed by the material in the case of impact, the test was carried out with a 6 J Charpy pendulum (Metrotec, San Sebastian, Spain). Five measurements have were taken, with bending type specimens and without notch. The parameters followed in the test are those defined in the ISO 179-1:2010 standard [70].

Finally, the materials surface hardness was measured. Five measurements were taken at different points of the surface of several specimens. A Shore D scale hardness tester, JBA S.A. model 673-D (Instruments JBot, S.A. Cabrils, Barcelona, Spain), was used. The parameters followed in the test are those defined in the ISO 868:2003 standard [71].

#### 2.4.2. Thermo-Mechanical Properties

To evaluate the effects of temperature on the mechanical properties of the materials, a study of softening temperature (VICAT) and heat deflection temperature (HDT) was carried out. Both tests were carried out with a Metrotec model (San Sebastian, Spain). The parameters for the VICAT test were those of the B50 method, as defined in the ISO 306 standard [72]. The standard establishes a load of 50 N and raises the temperature to 50 C h^−1^ until the penetrator penetrates 1 mm into the sample. For the HDT test, procedure B of the ISO 75-1:2013 standard [73] was used. The standard determines the use of 0.45 MPa load, using a weight of 74 g, and carrying out the test at 120 °C min^−1^ until reaching 0.31 mm of deformation.

#### 2.4.3. Thermal Properties

The materials’ miscibility was evaluated through the analysis of their thermal properties, using the differential scanning calorimetry technique (DSC). The thermoanalytical tests were carried out in a Mettler Toledo 882e (Mettler Toledo S.A.E., Barcelona, Spain). The weight range of the samples was 5–10 mg, with an initial heating program from −50 to 180 °C to remove the thermal history, followed by a cooling program from 180 to −50 °C, and finally a second heating program from −50 to 220 °C, at a heating rate of 10 °C min^−1^ in a nitrogen atmosphere with a nitrogen flux of 66 mL min^−1^. With this technique, the glass transition temperature (T_g_), cold crystallization peak (T_cc_), and the melt peak temperature (T_m_) of the PLA blends formulations were identified. The degree of crystallization (X%) was calculated with the formula by Equation (2).
(2)$$X_{C}\left(\%\right)=100\times \frac{\left|\Delta H_{CC}+\Delta H_{m}\right|}{\Delta H_{m}\left(100\%\right)}\times \frac{1}{W_{PLA}}$$
where $\Delta H_{CC}\;$ is the cold crystallization enthalpy, $\Delta H_{m}$ is the melt enthalpy, $\Delta H_{m}$ (100%) is the theorical melt enthalpy for a fully crystalline PLA structure (93 J g^−1^) [74], and WPLA is the PLA weight fraction.

Samples were tested by thermogravimetric analysis. The thermogravimetric analysis (TGA) was carried out in a TGA PT1000 equipment from Linseis Inc. (Selb, Germany). The analyzed samples, with an average weight between 15 and 20 mg, were subjected to a temperature program from 30 °C to 700 °C at a speed of 10 °C min^−1^, with a constant nitrogen flow of 30 mL min^−1^.

#### 2.4.4. Microscopic Characterization

The morphology of the blends was analyzed with a ZEISS ULTRA microscope from Oxford Instruments (Oxfordshire, UK) using a 1.5 KV accelerating voltage. The analyzed samples were the broken impact test specimens, which were sputtered with a thin layer of platinum, using a Leica Microsystems (Buffalo Grove, IL, USA) EM MED0200 high vacuum sputter coater.

#### 2.4.5. Disintegration under Composting Conditions

The study of the materials’ disintegration behavior, under composting conditions, was carried out according to the ISO 20200:2015 standard [75]. The tested materials were cut into fragments of 25 × 25 mm, and eight fragments of each type of material were made. The samples were buried in a compost reactor. The degree of disintegration (*D*) was calculated in percentage with the following equation:(3)$$D=\frac{m_{i}-m_{r}}{m_{i}}$$
where *m_i_* is the initial dry mass of the test material and *m_r_* is the dry mass of the residual test material recovered from the sieving.

## 3. Results

### 3.1. Optimization of the Epoxidation Process Conditions of Epoxidized Corn Oil (ECO)

The epoxidation of corn oil was performed with a peroxide-to-corn oil molar ratio of 3:1, according to previous experience in the epoxidation process of other vegetable oils [76]. The epoxidation process was confirmed by analyzing the oxirane oxygen number and iodine number. Figure 2a shows how the conversion of epoxidizable double bonds into oxirane rings takes place in the epoxidation process, as it was observed that the value of oxirane oxygen before the process is 0 and rapidly increases during the first hours due to the wide availability of double bonds, and from the fourth hour, it slows down as most of the double bonds have already reacted.

At the end of the process, at 8 h, an oxirane oxygen index of 5.82 was obtained. Comparing this result with the theoretical rate, obtained from the amount of fatty acids in the corn oil, a conversion of 85.9% was achieved. In addition, the conversion of double bonds represented by the iodine index rapidly decreased during the first 2 h from values close to 140 until the end at 1.5 after 8 h of treatment, which reflects a greater decrease in double bonds than that reflected by the oxygen oxirane. This is due to reactions parallel to the conversion of double bonds into epoxy groups [56]. Analyzing the data obtained in the epoxidation process, it is possible to reduce the epoxidation process to 6 h, since the results of the conversion of double bonds into oxirane rings are obtained at 8 h.

### 3.2. Synthesis of Maleinized Corn Oil (MCO)

Figure 2b shows the evolution of the acid number values at the end of each one of the stages at different temperatures, namely 180, 200, and 220 °C. Initially, the AV is 0.15 mg KOH g^−1^ and at the end of the first hour at 180 °C, it is observed that it increases up to 24.4 mg KOH g^−1^, indicating that the maleinization reaction is occurring. After the second hour, at 200 °C, another large increase is observed, reaching an AV of 87.8 mg KOH g^−1^, and finally, after the last stage, at 220 °C, the AV reaches 109.2 mg KOH g^−1^. These results are in total agreement with values obtained by A. Perez-Nakai et al. with an AV of 105 for the maleinized hemp seed and 130 for the maleinized Brazil nut at the end of the epoxidation process [67]. In addition, Quiles Carrillo et al. indicated that commercial-grade maleinized linseed oil has an AV of between 105 and 130 mg KOH g^−1^ [77].

### 3.3. Mechanical Properties

PLA is a brittle material, and with the intention of improving this characteristic, mixtures were made with PHB. Moreover, different percentages of MCO and ECO were used as plasticizers and compatibilizers. The tensile strength, elongation at break and impact absorbed energy values of the studied materials are represented in Figure 3 and the results obtained in the flexural characterization and Young’s modulus and toughness modulus are represented in Table 2. It is observed that, when mixing 25% of PHB with PLA, the strength properties such as tensile and flexural strength, Young’s modulus, and flexural modulus are considerably reduced and ductile properties such as elongation at break and impact absorbed energy are improved [78]. By incorporating ECO in the PLA–PHB blend, the tensile and flexural strength decrease slightly with respect to the PLA–PHB blend and the ductile properties increase as the amount of ECO in the system increases, achieving an elongation at break of 127% when incorporating 10 phr of ECO, which is 450% more elongation compared to the PLA–PHB blend and 810% more compared to PLA. The impact absorbed energy and toughness modulus also increases when incorporating ECO, achieving the best result with 10 phr of ECO. The behavior of the materials when incorporating MCO is similar, however, the tensile strength value decreases to a lesser extent when incorporating ECO, and it remains between 30 and 33 MPa with amounts lower than 7.5 phr of MCO and the flexural strength values are also slightly higher when incorporating ECO. In addition, it is also observed that the ductile properties increase only up to 5 phr of MCO, and at higher amounts, both the elongation at break and the impact absorbed energy decrease. The B-5MCO material has similar mechanical properties to B-10ECO with an increase in elongation at break of approximately 460% more than PLA–PHB blends and 830% more than PLA.

Both plasticizers achieved an increase in the ductility of the initial formulation, as can be seen in the values of the elongation at break, toughness modulus, and impact absorbed energy. In the case of the tensile strength, a variation was observed, with the increasing percentage of plasticizer in the formulation, decreasing tensile strength values were obtained. The values of Young’s modulus obtained a variation, although in both cases, with values above the initial formulation. Additionally, both plasticizers managed to reduce the maximum flexural strength of the initial formulation, in such a way that the higher the plasticizer percentage, the lower the flexural strength values. In the case of flexural modulus, both plasticizers increased these values, except for the formulation with 10 wt% of MCO.

### 3.4. Thermo-Mechanical Properties

Regarding the thermo-mechanical properties, as shown in Table 2, the incorporation of PHB causes a decrease in both the Vicat softening temperature (VST) and heat deflection temperature (HDT), up to 53.8 and 52.5 °C, respectively, with respect to PLA (58.5 °C VST and 55.8 °C HDT). When ECO and MCO are added to blend PLA–PHB, it is observed that the HDT and VST values increase slightly. This is ascribed to the fact that the incorporation of the biomaterial and the plasticizers reduces the glass transition temperature, as shown below in the DSC section, and thus the deformation under high temperatures. Finally, in the VST results, it is observed that the incorporation of biomaterials and plasticizers reduces the VST values obtained by PLA. The incorporation of these materials means that the hardness is affected when PLA is raised to temperatures close to the glass transition temperature.

### 3.5. Thermal Properties

The thermal stability of the PLA, PLA–PHB blend, and plasticized formulations with different contents of ECO and MCO was assessed by thermogravimetry analysis (TGA). Table 3 shows some thermal parameters such as the degradation onset temperature (T5%), which indicates the temperature at which a 5% weight loss occurs, the maximum degradation rate (T_max_), which corresponds to the peak of the first derivative curve, and the endset (T90%), which indicates the temperature at which a 90% weight loss occurs. Neat PLA possesses a thermal stability value with a T5% of 311.3 °C, a T_max_ of 330.5 °C, and a T90% of 342.6 °C. However, by adding PHB to PLA, the T_max_ value increases considerably (by approximately 20 °C), whilst the addition of certain amounts of ECO or MCO to blend (B) formulations causes a slight decrease in the T5%, T90%, and T_max_ values. In both modified oils, these values are lower when the quantity is larger. These values are lower for the formulations with MCO. For example, the addition of 10 phr of ECO (B-10ECO) results in a T5% reduction of 15.5 °C, while with 10 phr of MCO (B-10MCO), the reduction is 21.3 °C with respect to the B formulation value (286.1 °C). Regarding the T_max_ temperature, it can be observed that both plasticizers lead to a decrease in temperature, obtaining the lowest values at approximately 20 °C and 26 °C lower for B-10ECO and B-10MCO, respectively. A similar tendency was observed by Garcia-Campo et al. [79] for PLA/PHB/PCL blends compatibilized with epoxidized soybean oil (ELO). The addition of ELO into the formulations resulted in a reduction of T5% by 19.1 °C and T_max_ by 20 °C. For the specific case of maleinized oils, the work of Perez-Nakai et al. [67] shows how the addition of 10 phr of maleinized hemp seed oil (MHO) decreases the T_max_ by 10 °C with respect to the value of neat PLA.

Regarding to the DSC results, Table 3 also shows the main thermal properties and Figure 4 shows the calorimetric graphs of samples B and B-5MCO. The effect of the addition of PHB to the PLA matrix reduces the glass transition temperature of PLA to a value of 56.1 °C (B formulation). As can be seen, both modified oils have a direct effect on some thermal properties of PLA. Unplasticized PLA has a glass transition temperature (T_g_) of 61.3 °C, a cold crystallization temperature (T_cc_) of 121.9 °C, and a melting temperature (T_m_) of 151.2 °C. Both plasticizers—ECO and MCO—decrease the Tg of PLA, which indicates an increase in the mobility of the polymer chains at lower temperatures, evidencing the plasticizing effect of both modified oils [80]. However, the greatest plasticizing effect is shown by the formulations with MCO, obtaining a decrease in T_g_ of up to almost 10 °C for the specific case of B-2.5MCO and in reference to the formulation B. The same plasticizing effect was observed by Dominguez-Candela et al. [81], who employed epoxidized chia oil in PLA. Regarding to the Tm, no significant variations were observed in Tm for the formulation with ECO. However, the formulations with MCO shows a slight decrease (of approximately 2–3 °C).

With respect to crystallinity, it is observed that PLA without plasticizing has a crystallinity of 7.1%, whereas as PHB is added, the PLA is totally amorphous. The PHB molecular chains intercalated between the PLA chains do not allow the PLA chains to be ordered. The incorporation of ECO or MCO shows a slight increase, however, the values are negligible. The T_cc_ peak is observed at higher temperatures with the incorporation of ECO or MCO. In fact, the T_cc_ peak temperature increases from 121.9 °C (neat PLA) to 127.1 °C for the B-7.5ECO formulation. This is due to the steric impediment exerted by the ECO on the PLA chains.

### 3.6. Field Emission Scanning Electron Microscopy (FESEM)

In this characterization technique, the breaking surface of the specimens was observed, in this case, as those obtained in the impact test. The PLA morphology, as shown in Figure 5a and Figure 6a, showed a typical surface of brittle materials with a smooth fracture surface and angular fracture flanks. When making the PLA blend with 25% PHB, as shown in Figure 5b, the morphology of the fracture surface was completely modified since it is no longer smooth, and the high roughness and rounded fracture flanks characteristic of ductile materials are observed. In addition, in Figure 6b, made at higher magnifications, a blend without gaps is observed due to the low miscibility and with the PLA acting as a matrix and the dispersed domains of PHB [82]. The samples of blends that contain ECO can be seen in Figure 5c–g, where the incorporation of ECO is progressive, and the fracture morphology of all of them is similar to the PLA–PHB blend with high roughness and rounded fracture flanks. In addition, the appearance of holes and cavities is also observed due to some type of plasticizer saturation for ECO contents and above on the blend (Figure 5e) and these holes increase in number as the amount of ECO in the blend increases up to 10 phr (Figure 5g) [83]. On the other hand, in Figure 6c–e, this effect is observed at higher magnifications, and, in addition, it allows us to appreciate that the gaps due to the accumulation of oil are mainly generated in the PHB domains.

The morphology of the fracture surface of the blends containing MCO can be seen in Figure 5h–l and Figure 6f–h. As occurs in the samples with ECO, the fracture surface remains smooth with rounded ridges, typical of ductile materials. Additionally, the voids also increase due to oil saturation when the amount of oil in the matrix increases, however, the first voids are observed at smaller amounts of MCO, starting at 2.5 phr (Figure 5i) and considerably increase these gaps up to 10 phr of MCO. These voids or cavities negatively affect the ductile properties of the material, as seen previously. When observing the images at 10,000× magnification (Figure 6f–h), it can be seen that in addition to the plasticizing effect, it also has a compatibilizing effect between PLA and PHB, since the dispersed domains or dispersed phase increase in number and represent more than 25% of the images, which possibly, due to these domains, can be a PLA–PHB blend thanks to the influence of the MCO.

As was verified by the mechanical properties of the different blends, the saturation of the oil negatively affects the ductile properties of the blends. In the PLA–PHB–ECO system, the sample with the best ductile behavior is the sample with 10 phr of ECO. However, in the PLA–PHB–MCO system, saturation occurs at lower amounts of oil, at approximately 5 phr of MCO, which has a similar morphology to sample B-10ECO. Therefore, to obtain the same plasticizing effect, it is achieved with less amount of MCO than ECO.

### 3.7. Disintegration under Composting Conditions

Finally, to evaluate the disintegration behavior of the developed materials, they were buried in a composting reactor carried out at a laboratory scale. The variation of the mass loss of the different materials under composting conditions can be seen in Figure 7. The disintegration of the PLA–PHB blends with ECO and MLO is characterized by presenting an initial induction period, where mass loss barely occurs, at approximately 7 days for PLA–PHB–ECO materials and 10 days for PLA–PHB–MCO materials. After this initial state, the disintegration of the different materials accelerates. In the case of the disintegration of the PLA–PHB blend, it does not occur until reaching the first 15 days, at which time the disintegration accelerates and disintegrates until reaching 80% at the end of the test. The incorporation of the modified corn oils into the blend disrupts the disintegration of the PLA–PHB blend. In the case of the incorporation of the MCO, it is observed that at higher percentages, the disintegration of the blend slows down, reaching between 50 and 60% loss of mass with low percentages of MCO (less than 5%) and decreasing below 30% of mass loss with percentages higher than 5% of MCO at the end of the test. However, the incorporation of ECO to the blend generates a different behavior regarding disintegration under composting conditions, since the degree of disintegration at the end of the test decreases very clearly with any of the percentages of ECO used, reaching a loss of mass between 20 and 35%.

## 4. Conclusions

This work analyzes the usefulness of the use of modified—epoxidized (ECO) and maleinized (MCO)— corn oil as plasticizing and/or compatibilizing agents of the blend 75 wt% PLA–25 wt% PHB. Both chemical modifications of corn oil were carried out at the laboratory level and both processes were carried out correctly according to the results obtained in the analysis of the number of oxirane oxygen and iodine number for the ECO and according to the evolution of the acid number value for the MCO.

The incorporation of both modified oils, ECO and MCO, allows to obtain PLA–PHB formulations with better ductile properties, requiring less MCO to achieve the same plasticizing effect as with ECO. The plasticizing effect was observed by differential scanning calorimetry due to the decrease in the glass transition temperature (Tg), decreasing by approximately 2 °C when using ECO and between 4 and 8 °C when using MCO in the PLA–PHB blend. Additionally, the plasticizing effect is observed to a greater extent in the mechanical properties, since the elongation at break and the impact absorbed energy improve considerably when using the oils in the PLA–PHB formulations, increasing the elongation at break by more than 400% when using 10% ECO and 5% MCO. The FESEM analysis reveals a particular morphology of the samples plasticized with oil, presenting spherical cavities dispersed in the matrix and that increase with the increase in oil in the formulations. Finally, the oils delay the disintegration in composting conditions with respect to the PLA–PHB formulation, delaying said degradation to a greater extent than the ECO and it is also affected by the amount of oil used.

Therefore, although for this specific study, the MCO shows better plasticization and compatibilization results, two new oils with greater commercial potential were obtained by chemical modification. Compared to other oils obtained from cereals or seeds, CO presents a higher abundance and greater stability against oxidation, which makes it a very interesting raw material from the industrial point of view for applications in both bio-based and synthetic polymers.

## Figures and Tables

**Figure 1 polymers-14-04205-f001:**
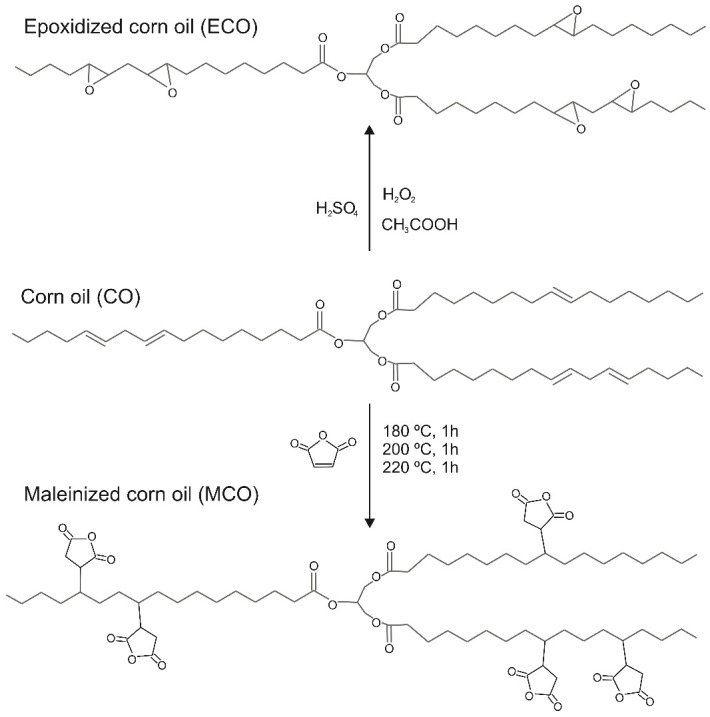
Schematic representation of the corn oil epoxidation and maleinization (MCO) processes.

**Figure 2 polymers-14-04205-f002:**
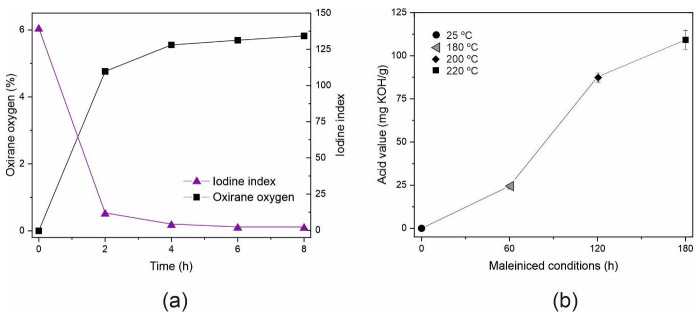
(**a**) Evolution of the oxirane oxygen index and iodine index in the corn oil epoxidation process and (**b**) evolution of the acid value in the different stages of the corn oil epoxidation process.

**Figure 3 polymers-14-04205-f003:**
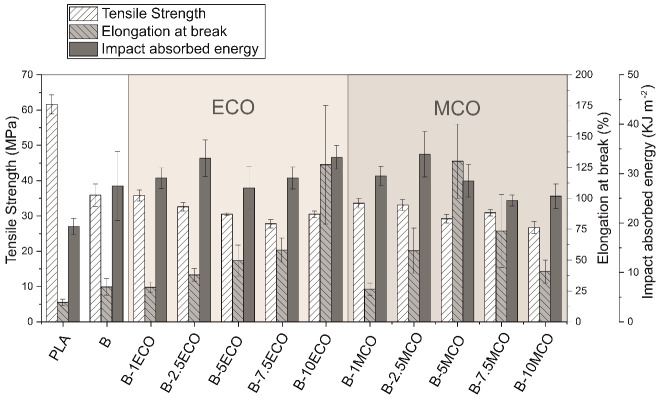
Effect of epoxidized and maleinized corn oil content on the tensile strength, elongation at break and impact energy of the PLA–PHB blend.

**Figure 4 polymers-14-04205-f004:**
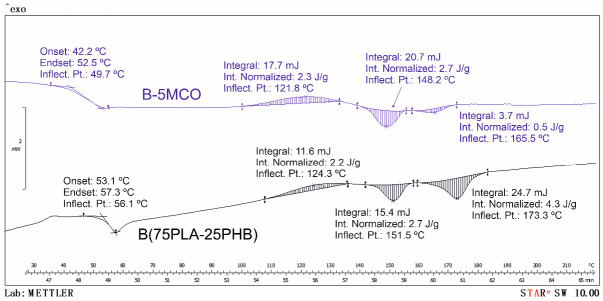
Calorimetric graphs of samples B (PLA–PHB) and B-5MCO.

**Figure 5 polymers-14-04205-f005:**
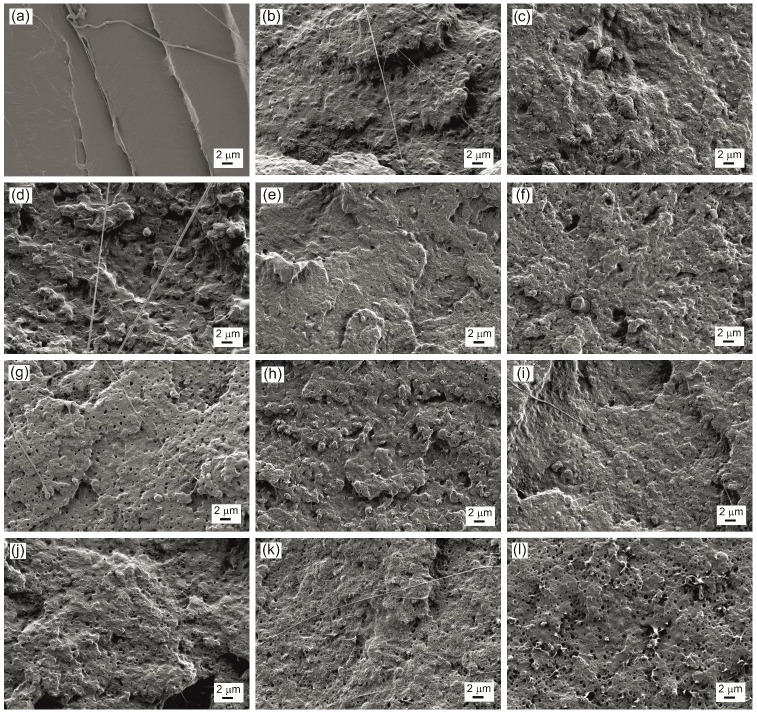
FESEM images at 2500× of fracture surface of (**a**) PLA; (**b**) PLA–PHB blend (B); (**c**) B-1ECO; (**d**) B-2.5ECO; (**e**) B-5ECO; (**f**) B-7.5ECO; (**g**) B-10ECO; (**h**) B-1MCO; (**i**) B-2.5MCO (**j**) B-5MCO; (**k**) B-7.5MCO; and (**l**) B-10MCO.

**Figure 6 polymers-14-04205-f006:**
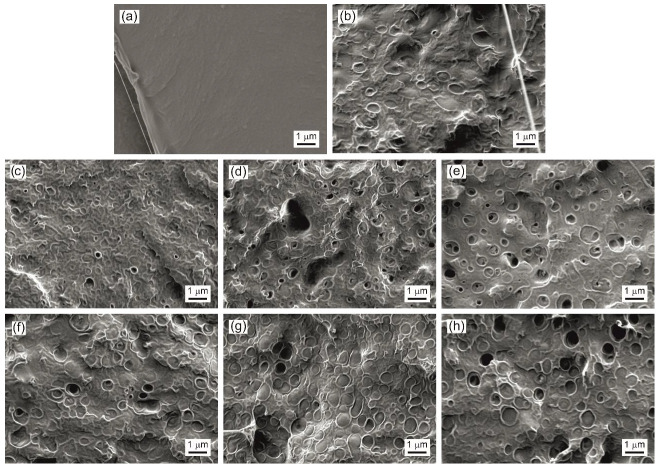
FESEM images at 10,000× of fracture surface of (**a**) PLA; (**b**) PLA–PHB blend (B); (**c**) B-5ECO; (**d**) B-7.5ECO; (**e**) B-10ECO; (**f**) B-5MCO; (**g**) B-7.5MCO; and (**h**) B-10MCO.

**Figure 7 polymers-14-04205-f007:**
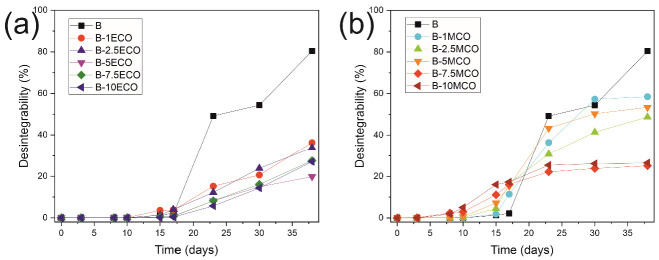
Degree of disintegration under composting conditions expressed as the weight loss as a function of time of (**a**) PLA–PHB blend and plasticized blend with different ECO content and (**b**) PLA–PHB blend and plasticized blend with different MCO content.

**Table 1 polymers-14-04205-t001:** Processed blends.

Code	PLA (wt%)	PHB (wt%)	ECO (phr *)	MCO (phr *)
PLA	100	-	-	-
B (blend)	75	25	-	-
B-1ECO	75	25	1	-
B-2.5ECO	75	25	2.5	-
B-5ECO	75	25	5	-
B-7.5ECO	75	25	7.5	-
B-10ECO	75	25	10	-
B-1MCO	75	25	-	1
B-2.5MCO	75	25	-	2.5
B-5MCO	75	25	-	5
B-7.5MCO	75	25	-	7.5
B-10MCO	75	25	-	10

* phr (per hundred resin) represents de weighted parts of the oil particles added to one hundred weight parts of the base PLA–PHB blend.

**Table 2 polymers-14-04205-t002:** Summary of tensile and flexural properties and HDT/VST results of neat PLA, PLA–PHB blend and plasticized blends with a different ECO and MCO content.

Sample	Tensile Strength (MPa)	Young’s Modulus (MPa)	Elongation at Break (%)	Toughness Modulus (MJ/m^3^)	Impact Absorbed Energy (kJ/m^2^)	Flexural Strength (MPa)	Flexural Modulus (MPa)	HDT (°C)	VST (°C)
PLA	61.6 ± 2.7	3470 ± 17	15.7 ± 2.7	6.0 ± 0.2	19.3 ± 1.6	99.8 ± 6.5	3290 ± 84	55.8	58.5
B (blend)	35.9 ±3.2	533 ± 102	28.3 ± 6.7	5.5 ± 1.0	27.5 ± 7.0	68.7 ± 4.0	2041 ± 92	52.5	53.8
B-1ECO	35.8 ± 1.6	1038 ± 168	28.0 ± 4.0	5.4 ± 0.4	29.1 ± 2.1	64.9 ± 2.9	2323 ± 154	54.2	53.5
B-2.5ECO	32.6 ± 1.2	1521 ± 22	38.0 ± 5.1	8.5 ± 1.0	33.1 ± 3.7	60.2 ± 0.7	2198 ± 195	54.5	53.7
B-5ECO	30.5 ± 0.4	1131 ± 182	49.6 ± 12.7	11.3 ± 3.8	27.1 ± 4.3	54.0 ± 3.6	2257 ± 84	52.4	54.3
B-7.5ECO	27.8 ± 1.2	1668 ± 207	58.1 ± 9.9	11.1 ± 1.0	29.1 ± 2.2	51.0 ± 0.6	2233 ± 115	54.2	53.3
B-10ECO	20.5 ± 0.9	1707 ± 48	127.2 ± 48.1	29.9 ±4.7	33.3 ± 2.4	54.7 ± 0.6	2392 ± 64	53.8	54.0
B-1MCO	33.6 ± 1.3	1189 ± 115	26.4 ± 4.8	6.4 ± 0.8	29.6 ± 1.9	64.8 ± 3.1	2291 ± 50	55.5	53.6
B-2.5MCO	33.1 ± 1.6	1403 ± 237	57.6 ± 18.3	11.3 ± 3.6	34.0 ± 4.6	61.6 ± 1.5	2235 ± 91	54.4	53.3
B-5MCO	29.2 ± 1.2	1372 ± 213	130.0 ±30.0	30.3 ± 4.4	28.6 ± 3.3	59.7 ± 1.0	2274 ± 261	54.4	53.0
B-7.5MCO	30.9 ± 0.9	1645 ± 54	73.5 ± 29.6	6.8 ± 2.7	24.6 ± 1.1	54.2 ± 2.1	2216 ± 100	54.1	53.7
B-10MCO	26.7 ± 1.7	1267 ± 102	40.8 ± 9.3	6.0 ± 1.1	25.5 ± 2.5	49.5 ± 1.5	2011 ± 93	55.0	52.6

**Table 3 polymers-14-04205-t003:** Summary of the main thermal parameters of the neat PLA, PLA–PHB blend, and plasticized blend with different ECO and MCO contents.

Code	TGA Parameters			DSC Parameters						
	T_5%_ (°C)	T_max_ (°C)	T_90%_ (°C)	T_g_ (°C)	T_cc_ (°C)	ΔH_cc_ (J/g)	T_mPLA_ (°C)	ΔH_m_ (J/g)	Xc (%)	T_mPHB_ (°C)
PLA	311.3	330.5	342.6	61.3	121.9	9.1	151.2	15.7	7.1	-
B (blend)	286.4	349.1	359.4	56.1	124.3	2.2	151.5	2.7	0.7	173.3
B-1ECO	276.4	342.8	350.4	54.4	125.6	2.3	151.0	2.7	0.6	172.5
B-2.5ECO	278.1	339.1	349.6	54.4	125.1	2.6	150.6	3.8	1.7	171.7
B-5ECO	275.5	338.2	349.4	54.1	125.2	1.5	151.2	2.6	1.6	173.9
B-7.5ECO	275.5	338.2	347.6	54.2	127.1	1.2	150.3	2.5	1.9	172.7
B-10ECO	270.9	328.7	338.7	54.6	126.7	2.6	151.8	4.1	2.7	175.7
B-1MCO	271.0	327.2	347.9	52.1	122.8	1.7	149.6	1.4	0.3	169.3
B-2.5MCO	270.1	329.2	348.0	47.3	122.4	0.9	149.8	1.2	0.4	166.1
B-5MCO	269.2	325.6	346.8	49.7	121.8	2.3	148.2	2.7	0.5	165.5
B-7.5MCO	269.1	323.2	345.9	49.6	122.1	2.6	148.2	4.1	2.4	165.1
B-10MCO	265.1	322.8	345.3	48.2	121.6	1.6	148.8	2.4	1.6	159.6

## Data Availability

Not applicable.

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
