# Peer review of "Effect of Epoxidized and Maleinized Corn Oil on Properties of Polylactic Acid (PLA) and Polyhydroxybutyrate (PHB) Blend"

_polymers, 2022, doi:10.3390/polym14194205_

Round 1

Reviewer 1 Report

The manuscript, entitled "Effect of epoxidized and maleinized corn oil on properties of PLA" by Jaume et al. presents an original research work on improving the mechanical properties of brittle PLA through incorporating a second polymer and plasticizers. This work clearly addresses the need for high performance bio-degradable polymers to resolve the environmental challenges. I could see the value of this work, but the authors need to address these few points before the manuscript being read by the readers of MDPI Polymers.

Major points:

1. One issue with this work is the usage of abbreviations without explaining them first. For example, I would recommend the authors to use "polylactic acid (PLA)" in the title. Additionally, please use the full name of PHB in abstract. 

2. The authors give an exhaustive introduction, reviewing the previous work that other researchers have been done. However, I think the authors need to address how innovative/different this work is compared with other published works. As the authors mentioned, there have been plenty of research on polymer blends and plasticizers. This work needs a stronger motivation.

3. The authors briefly mentioned the scope of this work (Line 76-80), but did not explain why epoxidized/maleinized corn oils rather than plain corn oil. Is that because of the incompatibility, or the instability of oil? Explaining the reason for this process is essential.

4. This manuscript also needs a much stronger Conclusion. In this current version, the authors only briefly summarized the key findings (Line 397-417). The authors need to connect these findings with the motivation of this work: how this work contributes to the deeper understanding in the field; how this work benefits the bio-degradable materials industry and sustainable future.

5. Please present stress-strain data. Additionally, please integrate stress-strain curves to obtain toughness of the materials in (almost) steady state (low rate). This is a parameter combining modulus/strength and elongation, giving a clear comparison of toughness.

6. Please add at least one (or a few) DSC curves. The authors presented a lot of DSC data, but it is not very clear how these numbers are obtained. Having DSC curves will be helpful for readers understanding the thermal process.

Minor points:

Grammars. For examples, Line 75, "All from natural origins" This is not a complete sentence. Line 351-355, could the authors break this sentence into a few shorter sentences? This current one is so long and confusing.

Author Response

Thank you very much for your review. All suggestions have been considered and changes done to the manuscript have been emphasized in yellow, in order to facilitate its searching. I summarize the main changes that you have suggested:

Reviewer 1

The manuscript, entitled "Effect of epoxidized and maleinized corn oil on properties of PLA" by Jaume et al. presents an original research work on improving the mechanical properties of brittle PLA through incorporating a second polymer and plasticizers. This work clearly addresses the need for high performance bio-degradable polymers to resolve the environmental challenges. I could see the value of this work, but the authors need to address these few points before the manuscript being read by the readers of MDPI Polymers.

Major points:

  1. One issue with this work is the usage of abbreviations without explaining them first. For example, I would recommend the authors use "polylactic acid (PLA)" in the title. Additionally, please use the full name of PHB in abstract.

ANSWER

As suggested by reviewer 1, the title has been modified. Also, in order to provide greater clarity we have included “polylactic acid (PLA) and polyhydroxybutyrate (PHB) blend” in the title, for this reason, we do not use the full name of PHB in the abstract. The new title is:

“Effect of epoxidized and maleinized corn oil on properties of polylactic acid (PLA) and polyhydroxybutyrate (PHB) blend”.

  1. The authors give an exhaustive introduction, reviewing the previous work that other researchers have done. However, I think the authors need to address how innovative/different this work is compared with other published works. As the authors mentioned, there has been plenty of research on polymer blends and plasticizers. This work needs a stronger motivation.

ANSWER

Thanks for this comment. We have added more information to explain our work's innovation or difference. It is in lines 81-94:

“These oils have been modified from corn oil due to certain characteristics that make it interesting compared to other oils from cereals or seeds. Corn oil is interesting since it has an interesting lipid profile for functionalization and has the advantage, compared to other modificable oils, of being able to be produced in large quantities and in many countries of the world, is economic and very stable to oxidation. The good oxidative stability is related to the low content of α-linolenic acid, being in the case of corn oil between 0-2%, depends on the genotype [63]. 85% of the fatty acids contained in the oil are unsaturated, which means a significant amount of double bonds that allow its functionalization by chemical processes such as epoxidation and maleinization. Therefore, this work evaluates the potential of epoxidized corn oil (ECO) and maleinized corn oil (MCO) as a bio-based plasticizer to improve the ductile properties of PLA-PHB blends”.

  1. The authors briefly mentioned the scope of this work (Line 76-80), but did not explain why epoxidized/maleinized corn oils rather than plain corn oil. Is that because of the incompatibility, or the instability of oil? Explaining the reason for this process is essential.

ANSWER

We totally agree with this reviewer’s comment. We have included explaining why we used modified corn oil in the Introduction section. It is in lines 81-84:

“The processes of maleinization and epoxidation of vegetable oils provide greater reactivity to the oil and greater thermal stability with respect to unmodified oils. Therefore, the compatibility of the oils with PLA is improved and the plasticizing effect is increased[56,62]”

  1. This manuscript also needs a much stronger Conclusion. In this current version, the authors only briefly summarized the key findings (Line 397-417). The authors need to connect these findings with the motivation of this work: how this work contributes to the deeper understanding in the field; how this work benefits the bio-degradable materials industry and sustainable future.

ANSWER

Thanks for this comment, we have revised the Conclusion and added 436-441 lines:

 “Therefore, although for the specific study the MCO shows better plasticization and compatibilization results, two new oils with greater commercial potential have been obtained by chemical modification. Compared to other oils obtained from cereals or seeds, CO presents a higher abundance and greater stability against oxidation, which makes it a very interesting raw material from the industrial point of view for application in both bio-based and synthetic polymers”

  1. Please present stress-strain data. Additionally, please integrate stress-strain curves to obtain toughness of the materials in (almost) steady state (low rate). This is a parameter combining modulus/strength and elongation, giving a clear comparison of toughness.

ANSWER

Thanks for your suggestion. We have made the toughness modulus calculations and added them together with the obtained tensile data to table 2. Also, in the "mechanical properties" section we have added information about the added data.

  1. Please add at least one (or a few) DSC curves. The authors presented a lot of DSC data, but it is not very clear how these numbers are obtained. Having DSC curves will be helpful for readers understanding the thermal process.

ANSWER

As reviewer 1 suggests, we have added Figure 4 with calorimetric graphs of sample B (PLA-PHB blend) and sample B-5MCO. We have only added two DSC curves because if we add more curves the information is not clear enough and the exothermic and endothermic peaks are not clearly seen when changing the scale.

Minor points:

Grammars. For examples, Line 75, "All from natural origins" This is not a complete sentence. Line 351-355, could the authors break this sentence into a few shorter sentences? This current one is so long and confusing.

ANSWER

As reviewer 1 suggests, we have changed these grammar mistakes.

Reviewer 2 Report

This work studied the effct of epoxidized and maleinized corn oil on properties of polylactic acid (PLA), an interesting work with novelty. I suggest acceptance for publication after addressing the following minor revisions.

1.  Title. If possible, I suggest the authors to add the full name of PLA into the title.

2. Introduction section. Some importance references should be cited, such as 

1) Ke, W.; Li, X.; Miao, M.; Liu, B.; Zhang, X.; Liu, T. Fabrication and Properties of Electrospun and Electrosprayed Polyethylene Glycol/Polylactic Acid (PEG/PLA) Films. Coatings 2021, 11, 790. https://doi.org/10.3390/coatings11070790

2) Ruan G, Feng S S. Preparation and characterization of poly (lactic acid)–poly (ethylene glycol)–poly (lactic acid)(PLA–PEG–PLA) microspheres for controlled release of paclitaxel[J]. Biomaterials, 2003, 24(27): 5037-5044. https://doi.org/10.1016/S0142-9612(03)00419-8

3. Eq. (1), the multiplication sign should be modified to avoid the misunderstanding.

4. If possible, I suggest the authors to add the FTIR data to better show the evolution of PLA-PHB blend after addition of ECO or MCO.

Round 2

Reviewer 1 Report

The authors have done exhaustive revisions to improve the quality of the manuscript. I think this is suitable for being accepted by MDPI Polymers.